# Comparison of Two Cuff Inflation Protocols to Measure Arterial Occlusion Pressure in Males and Females

Pat R. Vehrs [1,*], Chase Blazzard [1], Hannah C. Hart [1], Nicole Kasper [1], Ryan Lacey [1], Daniela Lopez [1], Shay Richards [1] and Dennis L. Eggett [2]

[1] Department of Exercise Sciences, Brigham Young University, Provo, UT 84602, USA
[2] Department of Statistics, Brigham Young University, Provo, UT 84602, USA
* Correspondence: pat_vehrs@byu.edu

**Abstract:** We measured the arterial occlusion pressure (AOP) in the dominant (DOM) and non-dominant (NDOM) legs of males ($n = 20$) and females ($n = 20$), 19–26 years of age, using a continuous (CONT) and incremental (INCR) cuff inflation protocol. ANOVA revealed no significant differences in AOP within (<1 mmHg; $p > 0.493$) or between (<6 mmHg; $p > 0.418$) the DOM and NDOM legs in males or females with either CONT or INCR. There were no significant sex differences in AOP in the DOM or NDOM legs when using CONT (<3 mmHg; $p > 0.838$) or INCR (<3 mmHg; $p > 0.856$). Measures of AOP are highly reliable, as evidenced by correlation coefficients >0.96 and small mean differences (<1.5 mmHg) between repeated measures. The choice of which cuff inflation protocol to use is one of personal preference. The AOP is not always greater in the dominant or larger leg. Although mean differences in AOP between the two legs was small, actual differences of over 100 mmHg could lead to unsafe and ineffective cuff inflation pressures during BFR if AOP is measured in only one leg. Further investigation of factors that explain difference in AOP between legs and between males and females is warranted. To ensure safe and effective use of BFR during exercise, AOP of both limbs should be measured regularly.

**Keywords:** blood flow restriction; blood flow restriction exercise; resistance training

## 1. Introduction

The use of blood flow restriction (BFR) of the limbs during low-load resistance training [1–5] is beneficial during musculoskeletal rehabilitation following an injury or surgery when the use of the traditionally recommended higher training loads are not feasible [4,6–9]. The results of using BFR during resistance training has also increased its use among athletes [10–12] and the fitness industry. The use of BFR during resistance training partially restricts arterial blood flow into the muscle and occludes venous blood flow [13–16]. Although the restriction of arterial blood flow can be accomplished with the use of elastic wraps [17–19], the use of an inflatable cuff allows for better control of blood flow restriction. The use of a given absolute cuff pressure (e.g., 200 mmHg) for the left and right limbs or for different individuals can result in different levels of blood flow restriction between the limbs or between individuals [14,15,20–24]. Thus, the current recommendations are to use a percentage of the limb's arterial occlusion pressure (AOP) to restrict blood flow during exercise [5,25–27]. Blood flow restriction during low-load resistance training is effective when the cuff is inflated to 40% to 80% of the limb's AOP [5,25–27]. Thus, the accurate measurement of the limbs AOP is essential to the safe and effective use of BFR during resistance training.

Studies measuring AOP have used two different protocols to inflate the occlusion cuff. One protocol increases the pressure of the occlusion cuff incrementally [28–31]. This involves inflating the cuff to 50 mmHg for 30 s, then deflating the cuff for 10 s, then inflating the cuff in increments of 30 mmHg for 30 s followed by deflating the cuff for 10 s until

arterial blood flow is no longer detected. The cuff is then inflated for 30 s to a pressure that is 10 mmHg less than the pressure at which arterial blood flow was no longer detected and then deflated for 10 s. This is repeated by inflating the cuff for 30 s to a pressure that is 10 mmHg less than the previous pressure, then deflated for 10 s, until arterial flow is again detected. The cuff is then slowly inflated to the point of arterial occlusion. The second protocol increases the pressure of the occlusion cuff continuously [32–44]. This involves inflating the cuff to 50 mmHg then gradually increasing the pressure of the cuff at a specific rate (e.g., 10 mmHg every 10 s) until arterial blood flow is no longer detected. When using either protocol, the cuff pressure at which arterial blood flow is no longer detected is recorded as the AOP. Although previous studies have used either of the two cuff inflation protocols, we are not aware of any study that has compared the AOP when using both cuff inflation protocols.

Although previous studies have included male and female participants [28,30,31,33–35, 38,39,41,43–45], only a few studies report sex differences in AOP. Jessee et al. [36] reported that the sex differences in AOP of the brachial artery of the right arm using three different occlusion cuff sizes (widths) were small and of no practical significance. Mouser et al. [40] also reported significant sex differences in AOP of 12 mmHg of the brachial artery of the right arm. Tafuna'i et al. [32] recently reported large and significant sex differences in the AOP of the superficial femoral artery in both the dominant and non-dominant legs.

Almost all studies have measured AOP in only one arm or one leg. Studies typically identify the limb of interest as the right or left limb and only a few studies identify the limb as the "dominant" arm [30] or leg [32,45]. In studies that have reported measurements or interventions in both limbs, differences in AOP between limbs have not been reported [6,28,46–50]. Two recent studies that have reported the AOP of both limbs. Evins et al. [33] reported a non-significant mean difference of less than 1 mmHg (95% CI = −9.8 to 8.5) in the limb occlusion pressure between the right and left legs. To the contrary, Tafuna'i et al. [32] reported a large and significant difference in the AOP between the dominant and non-dominant legs of males ($21 \pm 28$ mmHg; $p = 0.009$) but not of females ($13 \pm 27$ mmHg; $p = 0.053$). Since differences in the AOP between individuals is attributed primarily to differences in limb circumference [5,8,25,26,32,51], one might presume that differences in AOP between the two limbs within individuals may also be due to differences in limb circumference. Nevertheless, there were large differences in AOP between legs reported by Tafuna'i et al. [32] despite an average differences in leg circumference of less than 1 cm. In addition, AOP was greater in the smaller leg in 40% of the participants. Thus, differences in AOP between limbs may be due to other factors, such as differences in the cuff bladder position between the two limbs [52].

Few studies have reported the reliability of AOP measurements. A recent study reported high interrater (ICC = 0.894 to 0.984) and test-retest reliabilities (ICC = 0.737 to 0.985) of the brachial artery AOP [30]. Other studies have reported acceptable repeatability of AOP measurements of the brachial artery [29] and the femoral artery [45]. Ingram et al. [35] reported a time effect of measuring brachial artery AOP over four measurements over two days but the mean difference between measurements was relatively small and probably of little practical significance.

A recent review of the literature regarding BFR during exercise concluded a need for increased clarity of methodology used during BFR during exercise [53]. The primary purpose of this study was to compare the AOP of the superficial femoral artery (SFA) in the dominant and non-dominant legs in young healthy men and women when using a continuous and incremental cuff inflation protocol. The secondary purpose of this study was to evaluate the reliability of AOP measurements using the continuous and incremental cuff inflation protocols. We hypothesized no significant difference in AOP between the sexes or between the dominant and non-dominant legs, no significant differences in the AOP measured using continuous or incremental cuff inflation protocols, and reliable measurements of AOP.

## 2. Materials and Methods

This study compared the AOP of the SFA of the dominant and non-dominant legs as measured using two different cuff inflation protocols in males and females. To determine the variables that influence AOP, we also measured resting blood pressure, thigh circumference, thigh skinfold thickness, and thigh volume. Multiple measurements of AOP using both cuff inflation protocols permitted the evaluation of the reliability of AOP measurements. This study was reviewed and approved by the Institutional Review Board prior to the collection of any data.

### 2.1. Participants

A total of 40 (20 males, 20 females) physically active and apparently healthy adults, 19–26 years of age participated in this study. Interested participants with known risk factors for cardiovascular disease, one or more risk factors for thromboembolism, having been diagnosed or being treated for cardiovascular disease, renal disease, diabetes, or hypertension, currently pregnant or less than 6 months postpartum were excluded from participation in this study [36,38,41,48]. Participants were instructed to refrain from vigorous physical activity during the 24 h prior to their participation, consuming caffeine during the previous 8 h, and eating during the 2 h prior to their participation [38,41]. The methods, expectations, risks, and benefits of the study were explained to each participant, after which each participant voluntarily provided written informed consent. All measurements for each participant were completed in a single visit to the lab.

### 2.2. Procedures

The height (cm) and body mass (kg) or each participant was measured using a calibrated wall-mounted stadiometer scale (SECA Model 264; SECA, Chino, CA, USA) and a digital scale (Ohaus Model CD-33, Ohaus Corporation, Pine Brook, NJ, USA), respectively. Measured height and body mass were used to calculate body mass index (BMI; $kg/m^2$).

Participants self-reported leg dominance by responding to the question, "If you were to kick a ball, which leg would you use to kick the ball? [54]. Upper leg (thigh) volume ($m^3$) was calculated using the formula describing the volume of a truncated cone [55–59]. This involved partitioning the thigh into two segments using circumference measurements (cm) at three locations: the upper thigh (at the level of the gluteal fold), lower thigh (just above the proximal border of the patella) and mid-thigh (one-third the distance between the upper and lower circumference). The distances between the upper and mid circumferences and between the mid and lower circumference were measured in order to compute thigh volume. All circumferences were measured using a spring-loaded Gullick measuring tape. The thickness of the thigh skinfold was measured at the anterior midline of the thigh one-half the distance between the inguinal crease and the proximal border of the patella [60] using a calibrated Lange caliper (Santa Cruz, CA). Circumference and skinfold measurements were taken on the dominant and non-dominant legs in triplicate. If two of the measurements were the same, that value was used, otherwise all three measurements were averaged.

The participant then sat on a patient table in a semi-reclined position with the legs extended and supported with approximately 0° knee extension. A finger continuous noninvasive arterial pressure (CNAP) photoplethysmography unit (CNS Systems, Graz, Austria) was placed on the index and middle finger of the left hand to measure blood pressure and heart rate (HR) in real time. An inflatable calibration cuff was placed around the upper arm of the opposite arm. The participant rested in this position for 5 min after which the CNAP unit was turned on, allowed to calibrate, and resting HR, systolic (SBP), diastolic (DBP), and mean arterial pressures (MAP) were recorded.

As described in greater detail below, a Doppler ultrasound probe was placed on the randomly selected leg to determine the best position to record blood flow in the SFA. This position was marked with a pen. The circumference of the thigh 5 cm above the mark was measured as were the previous thigh circumferences. This circumference corresponded

to the center of the occlusion cuff that was to be placed around the thigh during the AOP measurements. This circumference was used in the analysis of data as a variable that could potentially influence the AOP. An uninflated SC10 Hokanson cuff (10 cm wide; 85 cm long) (Hokanson, Inc., Belleview, WA) was placed around the upper thigh and above the mark identifying the location to place the ultrasound probe for measuring AOP. The occlusion cuff was placed on the participant's thigh so the center of the cuff bladder was over the SFA [52].

### 2.3. Measurement of Arterial Occlusion Pressure

All measurements of AOP were performed using a Doppler probe (9 MHz; 55 mm) and GE ultrasound machine (GE LOGIQ, GE Healthcare). Inflation of the occlusion cuff was accomplished using a E-20 rapid cuff inflator (Hokanson, Bellevue, WA, USA) attached to the occlusion cuff. Presence of blood flow in the SFA was determined using the color flow and Doppler (pulse wave) modes. Angle of insonation of the ultrasound probe was maintained at 60°.

The AOP of the SFA was measured three times in each leg (6 times total) in a randomized order. The six measurements included the measurement of the AOP using a continuous cuff inflation protocol (CONT) in each leg, an incremental cuff inflation protocol (INCR) in each leg, and a third measurement of the AOP in each leg using either the CONT or INCR cuff inflation protocol. This allowed for the comparison of the AOP as measured using the CONT and INCR cuff inflation protocols in each leg, the comparison of the AOP between the two legs using the CONT and INCR cuff inflation protocols, and the reliability of AOP measurements using the CONT and INCR cuff inflation protocols. During the CONT protocol, the cuff was initially inflated to 50 mmHg, then gradually increased at a rate of 10 mmHg /10 s until arterial flow and pulse waves were no longer detectable. During the INCR protocol, the occlusion cuff was initially inflated to 50 mmHg for 30 s and then deflated for 10 s. The occlusion cuff was inflated incrementally with each subsequent inflation increasing by 30 mmHg for 30 s followed by deflation for 10 s until blood flow was occluded. Once blood flow was occluded, cuff pressure was decreased in increments of 10 mmHg for 30 s followed by deflation for 10 s until there was evidence of blood flow. Cuff pressure was then gradually increased at a rate of 10 mmHg/10 s until blood flow was no longer detected. In both cuff inflation protocols, AOP was defined as the lowest pressure at which arterial blood flow was occluded. After recording the AOP, the occlusion cuff and the CNAP cuffs were deflated. The investigator using the ultrasound Doppler probe to detect blood flow was blind to the pressure displayed on the cuff inflation system and the AOP recorded. The participant rested for 5 min [36,41,47] and the process was repeated in the next randomly selected leg and cuff inflation protocol. The CNAP was calibrated prior to each of the remaining AOP measurements.

### 2.4. Data Analysis

Differences in age, height, body mass, BMI, and resting blood pressure (SBP, DBP, MAP) and HR between males and females were determined using independent $t$-tests. Sex differences in thigh circumference, skinfold thickness, and thigh volume in the DOM and NDOM legs were also determined using independent $t$-tests. A familywise $p$-value of 0.05 was maintained using a pseudo Bonferroni adjustment to account for multiple comparisons. Based on the number of tests that we conducted, with the adjustment, $p$-values less than 0.01 were considered statistically significant.

The primary variable of interest in this study was the AOP measured in both legs using the CONT and INCR cuff inflation protocols. We compared sex differences in AOP in both legs using the CONT and INCR cuff inflation protocols, the AOP in each leg using the CONT and INCR cuff inflation protocols, the AOP between the two legs using the CONT and INCR cuff inflation protocols, and the reliability of the CONT and INCR cuff inflation protocols. Because there were multiple observations for each subject, we fit a mixed model analysis of variance blocking on subject to account for within- and

between-subject variability. To maintain a familywise *p*-value of 0.05, we used a pseudo Bonferroni adjustment to account for the multiple analyses, thus *p*-values less than 0.01 were considered statistically significant. To determine the variables that predict AOP, resting blood pressure (SBP, DBP, MAP), and thigh skinfold thickness, volume and circumference were included in a mixed model regression analysis. A sequential variable selection was used to determine which variable(s) best predicted AOP. In this analysis, *p*-values less than 0.05 were considered statistically significant. All analyses were performed using Statistical Analysis System version 9.4 (SAS Inc., Cary, NC, USA).

## 3. Results

On the average, males were taller, heavier, had a higher BMI and were slightly older than the female participants (Table 1). There were no sex differences in resting SBP, DBP, MAP and HR.

**Table 1.** Participant Characteristics.

|  | Males (*n* = 20) | Females (*n* = 20) | Difference | *p*-Value |
|---|---|---|---|---|
| Age (yrs) | 23.0 ± 1.5 | 21.6 ± 1.4 | 1.4 ± 0.5 * | 0.005 |
| Height (cm) | 179.3 ± 6.4 | 167.8 ± 7.5 | 11.5 ± 2.2 * | 0.001 |
| Body Mass (kg) | 82.1 ± 11.7 | 61.6 ± 6.7 | 20.4 ± 3.0 * | 0.001 |
| BMI (kg/m$^2$) | 25.6 ± 4.1 | 21.8 ± 1.9 | 3.7 ± 1.0 * | 0.001 |
| SBP (mmHg) | 116.3 ± 8.8 | 112.6 ± 7.8 | 3.6 ± 2.6 | 0.177 |
| DBP (mmHg) | 76.5 ± 6.8 | 75.2 ± 5.1 | 1.2 ± 1.9 | 0.514 |
| MAP (mmHg) | 90.1 ± 6.4 | 87.7 ± 5.2 | 2.4 ± 1.8 | 0.204 |
| Heart Rate (bpm) | 70.9 ± 8.5 | 73.1 ± 10.3 | 2.1 ± 2.9 | 0.477 |

Values are mean ± SD. BMI = body mass index, SBP = systolic blood pressure, DBP = diastolic blood pressure, MAP = mean arterial pressure. * = significant difference between males and females (*p*-value < pseudo Bonferroni adjusted family-wise *p*-value of 0.01).

Thigh volume and thigh skinfold thickness were not significantly different between males and females in either the DOM or NDOM leg (Table 2). Sex differences in thigh circumference approached significance in both the DOM and NDOM leg. There were no significant differences in thigh circumference, thigh skinfold thickness, or thigh volume between the DOM and NDOM legs in either males or females (*p* = 0.573–0.931).

**Table 2.** Leg Dimensions.

|  | Males (*n* = 20) | Females (*n* = 20) | Difference | *p*-Value |
|---|---|---|---|---|
| Thigh Skinfold (mm) |  |  |  |  |
| Dominant Leg | 23.3 ± 11.3 | 26.0 ± 5.7 | 2.7 ± 2.8 | 0.337 |
| Non-dominant Leg | 23.6 ± 10.6 | 25.7 ± 5.6 | 2.1 ± 2.7 | 0.434 |
| Difference | −0.30 ± 3.4 | 0.25 ± 1.7 | −0.4 ± 0.8 | 0.592 |
| Thigh Circumference (cm) |  |  |  |  |
| Dominant Leg | 59.6 ± 5.5 | 56.0 ± 2.9 | −3.7 ± 1.4 | 0.012 |
| Non-dominant Leg | 59.2 ± 5.2 | 55.6 ± 3.2 | −3.6 ± 1.4 | 0.012 |
| Difference | 0.43 ± 1.6 | 0.37 ± 0.9 | −0.06 ± 0.4 | 0.893 |
| Thigh Volume (m$^3$) |  |  |  |  |
| Dominant Leg | 0.236 ± 0.051 | 0.211 ± 0.034 | −0.024 ± 0.014 | 0.079 |
| Non-Dominant Leg | 0.227 ± 0.047 | 0.205 ± 0.031 | −0.022 ± 0.012 | 0.043 |
| Difference | 0.009 ± 0.022 | 0.006 ± 0.010 | −0.003 ± 0.007 | 0.684 |

Values are mean ± SD. No significant differences between males and females in the dominant and non-dominant legs in any of the three measures. No significant differences between the dominant and non-dominant legs in either males or females in any of the three measures (*p* value > pseudo Bonferroni adjusted family-wise *p* value of 0.01).

The analysis of variance revealed that there were no significant differences in the AOP between the DOM and NDOM legs in males or females with the CONT (*p* = 0.738, 0.431) or INCR (*p* = 0.654; 0.714) cuff inflation protocols, respectively (Table 3). Likewise, there were no significant differences in the AOP when measured with the two cuff inflation protocols in the DOM or NDOM legs in males or in females (Table 3). There were also no significant sex differences in the AOP in either the DOM or NDOM leg when using

the CONT ($p > 0.838$) or INCR ($p > 0.856$) cuff inflation protocol (Table 3). Because there were no statistically significant differences in the AOP between the DOM and NDOM legs or between males and females, we pooled data. There were no statistically significant differences in the AOP when measured using either cuff inflation protocol in either the DOM or NDOM legs or between the DOM and NDOM legs. Finally, after combining data from both legs, there were no statistically significant differences ($p = 0.990$) in the AOP when measured using the CONT and INCR cuff inflation protocols.

**Table 3.** Arterial Occlusion Pressure Using Two Cuff Inflation Protocols.

| | Continuous Cuff Inflation | Incremental Cuff Inflation | Difference | *p*-Value |
|---|---|---|---|---|
| MALES | | | | |
| Dominant Leg ($n = 20$) | $209.4 \pm 29.4$ | $208.5 \pm 27.1$ | $0.9 \pm 10.2$ | 0.682 |
| Non-dominant Leg ($n = 20$) | $206.2 \pm 31.5$ | $204.6 \pm 30.5$ | $1.6 \pm 8.2$ | 0.394 |
| Difference | $3.2 \pm 42.8$ | $3.9 \pm 38.2$ | | |
| Combined Legs ($n = 40$) | $207.8 \pm 30.1$ | $206.5 \pm 28.8$ | $1.3 \pm 9.1$ | 0.383 |
| FEMALES | | | | |
| Dominant Leg ($n = 20$) | $211.3 \pm 57.8$ | $210.5 \pm 53.8$ | $0.8 \pm 8.1$ | 0.643 |
| Non-dominant Leg ($n = 20$) | $203.5 \pm 48.3$ | $207.0 \pm 50.2$ | $-3.4 \pm 8.1$ | 0.073 |
| Difference | $7.8 \pm 43.4$ | $3.5 \pm 42.1$ | | |
| Combined Legs ($n = 40$) | $207.4 \pm 52.7$ | $208.7 \pm 51.4$ | $-1.3 \pm 8.3$ | 0.327 |
| SEX DIFFERENCES | | | | |
| Dominant Leg ($n = 20$) | $1.9 \pm 14.5$ | $2.0 \pm 13.5$ | | |
| Non-dominant Leg ($n = 20$) | $-2.6 \pm 12.9$ | $2.4 \pm 13.1$ | | |
| COMBINED MALES/FEMALES | | | | |
| Dominant Leg ($n = 40$) | $210.4 \pm 45.2$ | $209.5 \pm 42.2$ | $0.9 \pm 9.1$ | 0.534 |
| Non-Dominant Leg ($n = 40$) | $204.8 \pm 40.3$ | $205.8 \pm 41.1$ | $-0.9 \pm 8.4$ | 0.493 |
| Difference | $5.5 \pm 42.6$ | $3.7 \pm 39.7$ | | |
| Combined Legs ($n = 80$) | $207.6 \pm 42.6$ | $207.6 \pm 41.4$ | $-0.01 \pm 8.7$ | 0.990 |

Values are mean $\pm$ SD. No significant differences (adjusted $p > 0.01$) in all comparisons.

We considered multiple variables as predictors of AOP. The coefficients and significance of the predictor variables were calculated from the ANOVA results. The data analysis indicated that the coefficient for thigh circumference ($2.43 \pm 1.10$) was the only factor that significantly contributed ($p = 0.0278$) to AOP. Resting blood pressure (SBP, DBP, MAP), thigh skinfold thickness or thigh volume did not significantly contribute to AOP. After accounting for thigh circumference, no other factor contributed to AOP. There were no significance differences between the first and second measures of AOP using either the CONT or INCR cuff inflation protocols in either the DOM or NDOM legs in males or females (Table 4). The Pearson Correlation Coefficients between the first and second measurements of AOP using CONT or INCR cuff inflation protocol all exceeded 0.99.

**Table 4.** Repeated Measures of AOP Using Two Cuff Inflation Protocols.

| | Continuous Cuff Inflation | Incremental Cuff Inflation |
|---|---|---|
| MALES | | |
| First Measurement ($n = 20$) | $215.5 \pm 35.7$ | $198.1 \pm 22.2$ |
| Second Measurement ($n = 20$) | $214.6 \pm 33.6$ | $196.7 \pm 19.7$ |
| Difference | $0.9 \pm 5.4$ | $1.3 \pm 6.2$ |
| FEMALES | | |
| First Measurement ($n = 20$) | $207.5 \pm 51.4$ | $209.8 \pm 54.3$ |
| Second Measurement ($n = 20$) | $207.4 \pm 50.1$ | $210.6 \pm 58.2$ |
| Difference | $-0.4 \pm 7.9$ | $-0.8 \pm 11.8$ |
| COMBINED MALES/FEMALES | | |
| First Measurement ($n = 40$) | $211.3 \pm 43.7$ | $203.9 \pm 41.4$ |
| Second Measurement ($n = 40$) | $211.0 \pm 42.3$ | $203.6 \pm 43.5$ |
| Difference | $0.3 \pm 6.7$ | $0.3 \pm 9.4$ |

Values are mean $\pm$ SD. No significant differences ($p > 0.05$) between the first and second measures of AOP using either of the cuff inflation protocols (continuous; incremental) in males or females or when pooling male and female data.

## 4. Discussion

This paper adds to the current body of knowledge about using BFR during exercise in that we report, perhaps for the first time, agreement between two cuff inflation protocols used to measure AOP and that both protocols are highly reliable. We also report similarities in the AOP of the SFA between males and females and confirm results from previous studies that limb circumference is the primary determinant of AOP.

### 4.1. Differences in AOP with Different Cuff Inflation Protocols

Even though previous studies have used either a CONT [32–44] or INCR [28–31] cuff inflation protocol when measuring AOP, we are not aware of any studies that have compared measures of AOP when using the two cuff inflation protocols. In this study, there were no statistically significant differences in the AOP when measured using a CONT or INCR cuff inflation protocol in males or females or in the DOM or NDOM legs (Table 3). The AOP was higher in 40 (50%) of the 80 paired measurements when using the CONT cuff inflation protocol and was higher in 40 (50%) of the paired measurements when using the INCR cuff inflation protocol. When combining the data from males and females and both legs, the mean difference between the AOP when measured using a CONT and INCR cuff inflation protocol was nearly zero (Table 3). Of the 80 paired measurements of AOP using the two cuff inflation protocols, only 15 had a difference greater than 10 mmHg and the greatest difference in AOP when using the two cuff inflation protocols was 20 mmHg. Considering that the recommended cuff pressure when using BFR during exercise is 40% to 80% of the AOP, a difference in AOP of 20 mmHg between the two cuff inflation protocols would result in a difference of cuff pressure of up to 8 to 16 mmHg. If a cuff pressure near the middle of the recommended range is used during BFR during exercise, differences in the AOP when using the two cuff inflation protocols would not likely affect the safety of BFR during exercise or its effectiveness. Those who use BFR during exercise could therefore chose which cuff inflation protocol to use when measuring AOP. The CONT cuff inflation protocol requires less time whereas the INCR cuff inflation protocol provides some intermittent relief from the increasing pressure. For example, at the lowest and highest AOPs measured in this study (150 mmHg; 400 mmHg), the CONT and INCR cuff inflation protocols took approximately 2–6 min and 5–15 min, respectively. The time required to determine the AOP may be of particular importance when measuring AOP in the legs where arterial occlusion pressures can be quite high.

### 4.2. Limb Differences in Arterial Occlusion Pressure

In this study, there were no statistically significant differences in the AOP between the DOM and NDOM legs using either the CONT or INCR cuff inflation protocol in either males or females (Table 3). The non-significant differences in the AOP between legs is in agreement with a recently reported [33] average difference of less than 1 mmHg between legs. Our data and that of Evins et al. [33] is contrary to the results reported by Tafuna'i et al. [32] who reported large significant differences in AOP between legs. Our data indicating that AOP is closely related to size of the limb (i.e., circumference) rather than thigh volume, thigh skinfold thickness, and blood pressure is in agreement with that of Tafuna'i et al. [32]. Evin et al. [33] recently reported that 47% of the variance in limb occlusion pressure was attributed to between participants and 6% of the variance was attributed to between legs. Although limb circumference is a primary determinant of AOP, other factors may also contribute to differences in AOP between individuals (e.g., training and fitness status, limb composition, vascular health, blood pressure, time of day, age, and measurement and random error) and between limbs (i.e., differences in the position of the cuff bladder on the leg) [52].

Because there were no statistically significant differences in the AOP between the DOM and NDOM legs in both males and females in this study and that of Evin et al. [33], one might conclude that measuring the AOP in only one leg prior to the use of BFR during exercise is necessary. This would be a misinterpretation of the data. Although the

mean difference in the AOP between the DOM and NDOM legs was relatively small, the actual differences ranged from −107 mmHg to 124 mmHg using the CONT cuff inflation protocol and −107 mmHg to 110 mmHg using the INCR cuff inflation protocol. Using a recommended cuff pressure during BFR during exercise of 40% to 80% of the AOP measured in just one leg could result in an under- or over-inflation of the cuff of up to 88 mmHg when using the INCR cuff inflation protocol to measure AOP and 99 mmHg when using the CONT cuff inflation protocol to measure AOP.

In this study of 40 participants, the DOM leg was larger in 24 (60%) of the participants, the NDOM leg was larger in 13 (32.5%) of the participants, and the two legs were the same size in 3 of the participants. The larger leg had a higher AOP in 19 (47.5%) of the participants while the smaller leg had a higher AOP in 18 (45%) participants. The DOM leg had a higher AOP in 26 (65%) of the participants and the NDOM leg had a higher AOP is 14 (35%) of the participants. Thus, the DOM leg is not always the larger leg and the larger or DOM leg does not always have a higher AOP. This data is consistent with that recently reported by Tafuna'i et al. [32].

Measuring AOP in only one leg could result in cuff inflation pressures when using BFR during exercise that are unsafe or ineffective. Those who use BFR during exercise should not be naive of the potential differences in AOP between limbs. Our data suggest that AOP should be measured in each limb prior to the use of BFR during exercise.

### 4.3. Sex Differences in Arterial Occlusion Pressure

As the differences in AOP between individuals and between limbs can be attributed primarily to differences in limb circumference [14,51,61], sex differences in AOP may also be attributed, in part, to difference in limb circumference. In this study, although sex differences in thigh circumference approached significance (Table 2), the average differences in AOP between males and females in the DOM and NDOM legs using either cuff inflation protocol were small and non-significant (Table 3). Our data is contrary to the results of a recent study [32] that reported large significant differences in AOP between legs and between males and females despite small and insignificant differences in thigh circumferences. One possible explanation is that there is a wide variability in AOP for a given thigh circumference (Figure 1 in reference [32] Figure 1). In this study, and that of Tafuna'i et al. [32], after accounting for thigh circumference, other factors such as thigh skinfold thickness, thigh volume, SBP, DBP, MAP, age and sex were not significant independent predictors of AOP [32]. To the contrary, Jessee et al. [36] reported that after accounting for arm circumference, sex remained a significant independent predictor of AOP. One possible explanation for the differences in the results of this study and that of Jessee et al. [36] is the difference in the size of the limbs studied (i.e., legs vs. arms) and the magnitude of the AOP between the arms and legs.

### 4.4. Reliability of AOP Measurements

In this study, the average differences between the first and second measurement of AOP was less than 1 mmHg and 1.3 mmHg when using the CONT and INCR cuff inflation protocols, respectively (Table 4). The range of differences between the first and second measurement of AOP was less than 12 mmHg when using the CONT cuff inflation protocol and less than 22 mmHg when using the INCR cuff inflation protocol. This data, combined with high correlations (>0.99) indicate that sequential measurements of AOP are highly reliable.

In this study, we measured AOP in each leg using each cuff inflation protocol within a single visit to the laboratory. Thus, we did not assess day-to-day or administrator variability. The recent study by Evin et al. [33] reported that 47% of the variance in limb occlusion pressure was between participants, 18% was within participants between days, 6% was within participants between legs, and 28% was associated with random error. Due to a high relative error of 14.4 mmHg, Evin et al. [33] suggested that the limb occlusion pressure should be measured regularly to ensure effectiveness of BFR during exercise. This

concurs with the recommendation of another previous study that AOP should be measured regularly during training (as opposed to a single time point) since AOP may vary within and between days [8].

*4.5. Study Limitations*

This study is not without limitations. This study included untrained but physically active young adult coeds without known risk factors for cardiometabolic diseases. Therefore, the results of this study may not apply to all subgroups of the population. In this study, we used a research- clinical-grade cuff inflation system (Hokanson, Bellevue, WA, USA) and a 10 cm cuff for all measurements of AOP. Those who use BFR during exercise in a variety of settings likely use other cuff inflation systems, cuff sizes, or other methods to occlude blood flow.

*4.6. Directions for Future Studies*

Differences in measurements of AOP between limbs, between multiple measurements, between individuals, or when using different cuff inflation protocols, may be related to how the cuff is inflated. In this study, we used a clinical grade cuff inflation system (Hokanson E-20 rapid cuff inflator; Hokanson, Bellevue, WA, USA) which allows dependable and easily-controlled cuff inflations. Previous studies have also used this system [28,31,32,47,49,52]. Use of other methods to inflate the cuff, such as manual cuff inflation using a sphygmomanometer, may introduce error associated with the inflation of the cuff or reading the pressure from a dial.

To continue to expand our understanding of differences in AOP between limbs and individuals, future studies should include both male and female participants, report limb dominance and sex differences in AOP, and continue to measure factors that could contribute to differences in AOP, such as blood pressure, vessel diameter, limb circumference, limb volume, limb length, and subcutaneous fat thickness. These measurements can be taken each time AOP is measured over the course of repeated measurements on the same day or different days.

**5. Conclusions**

Important findings of this study were the agreement between the continuous and incremental cuff inflation protocols used to measure AOP in both legs and in males and females. Both cuff inflation protocols are highly reliable within a single visit, and the protocol of choice is a personal decision of the administrator or the participant. Even though mean differences between the DOM and NDOM legs were small and not statistically significant, actual differences of over 100 mmHg between the two legs could lead to the use of cuff inflation pressures during BFR that are either unsafe or ineffective if AOP is measured in only one leg. Our data affirms previous findings that AOP is not always greater in the dominant leg or the larger leg. To ensure safe and effective use of BFR during exercise, AOP of both limbs should be measured regularly. Our study confirms that leg circumference is the single most influential factor that contributes to AOP. Nevertheless, we found no significant sex differences in AOP in either the dominant or non-dominant leg despite near significant differences in leg circumference, suggesting the need for further investigation of factors that contribute to AOP.

**Author Contributions:** Conceptualization, P.R.V.; methodology, P.R.V., C.B., H.C.H., N.K., R.L., D.L. and S.R.; formal analysis, D.L.E. and P.R.V.; investigation, P.R.V., C.B., H.C.H., N.K., R.L., D.L. and S.R.; resources, P.R.V.; writing—original draft preparation, P.R.V.; writing—review and editing, D.L.E., C.B., H.C.H., N.K., R.L., D.L. and S.R.; project administration, P.R.V. All authors have read and agreed to the published version of the manuscript.

**Funding:** This research received no external funding.

**Institutional Review Board Statement:** The study was conducted according to the guidelines of the Declaration of Helsinki and approved by the Institutional Review Board of Brigham Young University (Protocol IRB2021-383; 3 January 2022).

**Informed Consent Statement:** Informed consent was obtained from all subjects involved in the study.

**Data Availability Statement:** The data presented in this study are available upon request from the corresponding author. The data are not publicly available.

**Conflicts of Interest:** The authors declare no conflict of interest.

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
