# Peer review of "Comparison of Two Cuff Inflation Protocols to Measure Arterial Occlusion Pressure in Males and Females"

_applsci, doi:10.3390/app13031438_

Round 1

Reviewer 1 Report

The authors have compared two common AOP measurement protocols, a continuous inflation and an intermittent inflation protocol. They found that there were no significant differences between the two in the resulting AOP measurements.

Firstly, the authors have done and excellent job in designing, carrying out, describing, and reporting on their study. This is a very well-written manuscript and I have only one comment that I would like to see addressed.

At the top of page 7, in the paragraph immediately prior to Table 4 (lines 309-311), the authors state, "The data analysis indicated that thigh circumference was the only factor that significantly contributed (p = 0.0278) to AOP. Resting blood pressure (SBP, DBP, MAP), thigh skinfold thickness or thigh volume did not significantly contribute to AOP." While ANOVA is a special case of regression analysis, many readers will not immediately make this connection. I think this section could do with a single added sentence stating that the coefficients and significance for each of the predictor variables were calculated from the ANOVA results. This warrants an additional note in that you technically have 6 predictor variables (thigh circumference, systolic pressure, diastolic pressure, mean arterial pressure, thigh skinfold thickness, and thigh volume) across 80 observations. This gives you ~13.3 observations per predictor, while other studies using multiple hierarchical regression (specifically Matt Jessee's paper) have far more observations per predictor. And we could get into the debate about numbers of observations per predictor ranging anywhere from 10 being good enough to a minimum of 50 being gold standard, but that would neither add significantly to this manuscript, and it might actually detract from it. Still, something to consider in future work. I would be interested to hear your thoughts as to why your predictors did not line up with other lower-body work (Jeremy Loenneke's 2015 paper, doi:10.1007/s00421-014-3030-7) looking at predictors of occlusion in a coed population.

Thank you for the opportunity to review this manuscript. It adds to the literature on BFR and, more importantly, it included an equal number of women and men, unlike just about every other study out there in our small field. Cheers.

Reviewer 3 Report

I commend the authors on this investigation.  There are a couple of areas that I believe would improve the quality of the manuscript and clarity for readers.

1. Line 140: recommend change of 180 degrees to 0 degrees

 2. Line 160-162: what led to use of the Hokanson system/cuffs? Are there any reports about how frequently this unit is used in performance/rehab settings?  If not, consider stating how many of the referenced sources utilized the cuff system used in this manuscript.

3. It would be informative to include the amount of time (range and average) it took to complete each of the testing methods utilized (CONT, INCR).

4. Confirm accuracy and completeness of references: #22, 32, and 60.

Author Response

We have addressed each of the Reviewer's comments below.

Comment. Line 140: recommend change of 180 degrees to 0 degrees

Response. 180 degrees has been changed to 0 degrees.

Comment. Line 160-162: what led to use of the Hokanson system/cuffs? Are there any reports about how frequently this unit is used in performance/rehab settings?  If not, consider stating how many of the referenced sources utilized the cuff system used in this manuscript.

Response. The Hokanson cuff inflation system is a clinical grade cuff inflation system that provides easy control of cuff inflation, dependable inflations, etc. Several studied have used this system and we have included some information in the manuscript (Limitations section) to address your concerns.

Comment. It would be informative to include the amount of time (range and average) it took to complete each of the testing methods utilized (CONT, INCR).

Response.  We have included information about the time required to measure AOP using each of the inflation protocols at the end of Section 4.1.

Comment. Confirm accuracy and completeness of references: #22, 32, and 60.

Response. We have checked these references. We corrected a typo in the list of authors of reference 22. Otherwise the references are correct. The issue number of the articles are included in our Endnote file but are not inserted in the MDPI style.

Round 2

Author Response

Thank you for your additional suggestions. We have responded to your requests in lines 211-215 and lines 321-323 of the revised manuscript.